# Models of similarity in complex networks

Sergey Shvydun

HSE University, Moscow, Russia
Delft University of Technology, Delft, The Netherlands

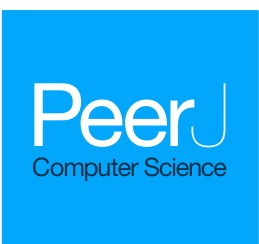

## ABSTRACT

The analysis of networks describing many social, economic, technological, biological and other systems has attracted a lot of attention last decades. Since most of these complex systems evolve over time, there is a need to investigate the changes, which appear in the system, in order to assess the sustainability of the network and to identify stable periods. In the literature, there have been developed a large number of models that measure the similarity among the networks. There also exist some surveys, which consider a limited number of similarity measures and then perform their correlation analysis, discuss their properties or assess their performances on synthetic benchmarks or real networks. The aim of the article is to extend these studies. The article considers 39 graph distance measures and compares them on simple graphs, random graph models and real networks. The author also evaluates the performance of the models in order to identify which of them can be applied to large networks. The results of the study reveal some important aspects of existing similarity models and provide a better understanding of their advantages and disadvantages. The major finding of the work is that many graph similarity measures of different nature are well correlated and that some comprehensive methods are well agreed with simple models. Such information can be used for the choice of appropriate similarity measure as well as for further development of new models for similarity assessment in network structures.

# INTRODUCTION
## Motivation of the study

Many real systems including social, financial, technological, biological, and informational can be represented as networks, where the elements of a system are nodes and interactions between elements are edges. In this regard, network theory is an important tool to model the structure and the dynamics of complex systems that may gain a comprehensive insight into many practical problems. However, since most of these systems evolve over time, there is a need to investigate the changes, which appear in the system, in order to assess the sustainability of the network and to understand how much the network has actually changed. Such information provides a better understanding of network evolution and it can be used to identify structural phase transitions in graphs or detect anomalies in real systems. It should be noted that graph similarity problem is also of great importance both for studying networks that change over time and for comparing systems of different

Corresponding author
Sergey Shvydun, shvydun@hse.ru

nature. Graph similarity models can be also used to graph matching or to compare real systems with theoretical models describing these systems.

## Research gaps

Unfortunately, evaluating graph similarity is an ill-defined problem. There is no universal definition for the similarity; consequently, there have been proposed diverse approaches to the problem. There is also no clear guideline on how to assess the performance of different methods. Therefore, there exist many models that measure the similarity between two networks. The simplest models rely on direct comparison of node or edge sets (*e.g.*, Jaccard index, graph edit distance, vertex/edge overlap, *etc.*). These models depend explicitly on the node labelling, thus, they are not invariant under graph isomorphism and cannot be applied to compare graphs with different nodes labels. Moreover, these measures are local and do not take into account the global structure of the network.

Several models aim to compare various characteristics that capture the global structure of the network. These characteristics may include the graph spectrum (*e.g.*, λ-distance, non-backtracking spectral distance, quantum Jensen–Shannon divergence), some empirical distributions (*e.g.*, communicability sequence entropy, distributional non-backtracking spectral distance), node affinities (*e.g.*, Deltacon), graph diffusion processes (*e.g.*, graph diffusion distance), or some other graph statistics. Most of these models are invariant to node labelling, hence, they can be used to compare graphs of different size and nature. Moreover, some of these measures satisfy all metric axioms.

There also exist hybrid models that take into account both local and global structure of the network and, in addition, consider the community structure or the centrality of nodes (*e.g.*, D-measure, LD-measure, SLRIC-similarity). However, some of these models do not satisfy the triangle inequality property while some other properties have not been studied in detail. Moreover, many similarity measures are designed for undirected unweighted networks, thus, they cannot be applied to directed or weighted graphs. Furthermore, some models cannot be applied to large networks due to their high computational complexity. Finally, the sensitivity of these models to various perturbations in the initial data has not been examined.

## Major finding

This work presents a survey of various models that measure the similarity between two networks. It considers 39 graph similarity measures, discusses their properties, compares their performance on real systems as well as on artificial graphs, and gives advice on their usage.

It is necessary to mention that there exist some reviews on graphs comparison. In *Soundarajan, Eliassi-Rad & Gallagher (2014)* the authors consider 20 similarity measures and perform their correlation analysis on networks from diverse domains. In *Donnat & Holmes (2018)*, the authors provide an overview of eight graph distances, discuss some structural changes that they are best able to capture and highlight their performance on both synthetic and real-data graphs. *Tantardini et al. (2019)* consider 12 distance measures and assess their performances both on synthetic benchmarks and on real-world multilayer

networks. In *Wills & Meyer (2020)*, the authors compare seven distance measures on random graphs and on some real networks and discuss their computational complexity. They also provide some recommendations on the choice of the appropriate distance measure and introduce their implementation in Python. In *Hartle et al. (2020)* the authors propose ensembles of random networks as benchmarks for network comparison methods. They also employ these benchmarks to examine some properties of 20 graph distances. All these studies are used as the backbone and we extend them.

The major contributions of this article can be summarized as follows:

- The article considers 39 graph similarity measures. To the best of the author's knowledge, there are no studies that have analyzed such a large number of models.
- The article incorporates existing approaches of graph similarity comparison and also introduces some novel approaches.
- The experiments demonstrate that models of different nature are surprisingly well correlated and that some comprehensive methods are agreed well to simple models.
- The author evaluates the runtime of graph similarity measures on random networks and reveal models that can be applied to large graphs.
- The author provides an implementation in Python of all graph similarity measures that are discussed in the article.

## Structure of the article

The article is organized as follows. In "Preliminaries and Properties of Distance Measures", the author recalls some definitions and discusses the theoretical properties that are usually examined for graph similarity measures. "Graph Distance Measures" considers various approaches of assessing graph similarity and provides their classification with respect to different criteria. "Validation" compares the models on artificial and real networks and evaluates their runtime. The final Section concludes.

## PRELIMINARIES

A network can be represented by a graph $G = (V, E)$, where $V = \{1, \ldots, n\}$ is a set of nodes, or vertices, and $E \subseteq V \times V$ is a set of edges (links), which connect the nodes. Two distinct nodes $i$ and $j$ are said to be neighbors, or adjacent nodes, if there is an edge $(i, j) \in E$ or $(j, i) \in E$ between them. Two nodes $i$ and $j$ are said to be connected if there exists a path from node $i$ to node $j$. The article considers undirected and directed graphs. All the graphs are simple and do not allow multiple edges (of the same direction) between a pair of nodes. For the latter, the existence of edge $(i, j)$ does not imply the existence of edge $(j, i)$. To describe a graph an adjacency matrix $A = [a_{ij}]$ is used where $a_{ij} = 1$ if there is edge $(i, j)$, and $a_{ij} = 0$ otherwise. If $E = \emptyset$, the graph is empty ($G_0$). Similarly, if the $E = V \times V$, the graph is a clique (complete graph, $K_n$). Denote by $N(i)$ a set of neighbors of node $i$ in graph $G$. Additionally, if connections between nodes are associated with some numerical values, representing the intensity of connections, the graph can be described by weighted adjacency matrix $W = [w_{ij}]$ that stores the weights of the edges.

Consider two arbitrary graphs $G_1 = (V_1, E_1)$ and $G_2 = (V_2, E_2)$. The union of nodes in graphs $G_1$ and $G_2$ is defined as $V = V_1 \cup V_2$, while the union of edges is defined as $E = E_1 \cup E_2$. The natural way to compute the similarity or dissimilarity between two graphs is to evaluate the distance between them. Denote by $d(G_1, G_2)$ a distance metric between graphs $G_1$ and $G_2$. In other words, the distance function is a mapping $d: (G_1, G_2) \to \mathbb{R}^+$. Usually, $d(G_1, G_2) = 0$ if graphs $G_1$ and $G_2$ are identical. On the contrary, large values of $d(G_1, G_2)$ correspond to a low similarity between two graphs. In some studies, the distance measure $d(G_1, G_2)$ is additionally transformed into the similarity measure $sim(G_1, G_2) \in [0, 1]$ where $sim(G_1, G_2) = 1$ for identical graphs and $sim(G_1, G_2) = 0$ for completely dissimilar graphs. In the literature, there have been proposed many ways of converting a distance metric. Some of them are discussed in *Koutra et al. (2011)*.

There are several ways how to measure the distance between graphs $G_1$ and $G_2$. Most approaches firstly collect a probability distribution $P$ from empirical data (*e.g.*, degree distribution, distance distribution, *etc.*) or a feature vector $\vec{x}$ (*e.g.*, average node degree, graph density, diameter, assortativity, eigenvalues, *etc.*) from each graph and then evaluate the distance between two graphs as the distance between these vectors or distributions.

To compare two vectors $\vec{x}$ and $\vec{y}$ the most prevalent ways of computing the distance are the following.

(1) *p-norm (Minkowski distance)*

The Minkowski distance is a metric that generalizes a wide range of distances (*Deza & Deza, 2009*). It is computed as

$$d(\vec{x}, \vec{y}) = \left( \sum_i |x_i - y_i|^p \right)^{1/p},$$

where $x_i$, $y_i$ are $i$-th coordinates of vectors $\vec{x}$ and $\vec{y}$, $p$ is a parameter. If $p = 1$, it is equal to L1-norm (Manhattan distance). If $p = 2$, it is L2-norm (Euclidean distance, Frobenius norm). The Minkowski distance is equivalent to Chebyshev distance if $p \to \infty$.

(2) *Canberra distance*

The Canberra distance is a weighted version of L1-norm (*Deza & Deza, 2009*). It is defined as

$$d(\vec{x}, \vec{y}) = \sum_i \frac{|x_i - y_i|}{|x_i| + |y_i|}.$$

There also exist different methods to compare two graphs $G_1$ and $G_2$ if they are characterized by probability distributions $P$ and $Q$. The most popular approaches are the following.

*(3) Jensen–Shannon (JS) divergence*

JS divergence $\mathcal{J}(P, Q)$ is one of the most common information-theoretic metrics satisfying the property of symmetry (*Cichocki & Amari, 2010*). It is defined as

$$\mathcal{J}(P, Q) = \frac{1}{2}\left(KL\left(P \,\middle\|\, \frac{P+Q}{2}\right) + KL\left(Q \,\middle\|\, \frac{P+Q}{2}\right)\right),$$

where $\dfrac{P+Q}{2}$ is the average distribution, $KL(P \parallel Q) = \sum p_i \log\left(\dfrac{p_i}{q_i}\right)$ is the Kullback–Leibler divergence for discrete probability distributions $P$ and $Q$.

Jensen–Shannon divergence can be also applied to compare more than two distributions. The equation for comparing $N$ distributions is the following

$$\mathcal{J}(P_1, \ldots, P_N) = \frac{1}{N}\sum_{i=1}^{N} KL(P_i \| P'),$$

where $P' = \dfrac{1}{N}\sum_{i=1}^{N} P_i$ is the average distribution. One should note that Jensen–Shannon divergence is bounded by $\log n$ ($0 \leq \mathcal{J}(P_1, P_2, \ldots, P_n) \leq \log n$) where $n$ is the number of distributions.

*(4) Earth mover's distance (EMD)*

The EMD is another widely used metric which compares two probability distributions $P$ and $Q$ (*Deza & Deza, 2009*). The main idea of EMD is that it measures the minimum effort required to transform one distribution to another ("to move earth from one pile to another"). In mathematics, this measure is also known as Wasserstein distance or Kantorovich–Rubinstein metric. The EMD can be solved as the optimal flow problem, *i.e.*,

$$EMD(P, Q) = \min_{F=\{f_{ij}\}} \sum_i \sum_j f_{ij} d_{ij},$$

where $f_{ij}$ is the flow between distributions $p_i$ and $q_i$ that minimizes EMD measure, $d_{ij}$ is the ground distance between $p_i$ and $q_i$.

More details on distance measures is provided in *Deza & Deza (2009)*. Before discussing various graph similarity measures some of their properties are discussed below.

## PROPERTIES OF DISTANCE MEASURES

The problem of measuring the similarity among the networks, intuitive at first sight, is not well defined. Indeed, there is no universal definition of the similarity or dissimilarity between two complex structures. Therefore, there have been proposed various concepts of similarity that capture different features of complex systems. One way to compare these models is to consider some rational properties that should be satisfied. The analysis of properties may provide a more detailed perception of the main features, advantages and disadvantages of the existing models and may justify the selection of a particular concept.

In geometry, the similarity can be treated as the distances in space, hence, it must follow the well-known metric axioms (*Deza & Deza, 2009*):

**1. *Non-negativity axiom*:** the distance between any two graphs is non-negative, *i.e.*,

$\forall G_1, G_2 \ \ d(G_1, G_2) \geq 0$

**2. *Identity of indiscernibles axiom*:** the dissimilarity of two graphs is zero if and only if they are identical, *i.e.*,

$d(G_1, G_2) = 0 \ \Leftrightarrow G_1 = G_2.$

**3. *Symmetry axiom*:** the dissimilarity of graph $G_1$ to $G_2$ is the same as the dissimilarity of $G_2$ to $G_1$, *i.e.*,

$d(G_1, G_2) = d(G_2, G_1).$

**4. *Triangle inequality*:** graphs $G_1$ and $G_2$ cannot be farther apart in dissimilarity space than the sum of their distances to any other graph $G_3$, *i.e.*,

$d(G_1, G_3) + d(G_3, G_2) \geq d(G_1, G_2).$

If the dissimilarity between two distinct graphs is zero, the function is a pseudo-metric. Similarly, the function is quasi-metric if it is not symmetric and semi-metric if it does not satisfy the triangle inequality. One should mention that many distances in real world do not satisfy these properties (*Cullinane, 2011*).

In many studies, the distance function is limited to some interval. In that case, *Koutra, Vogelstein & Faloutsos (2013)* introduces a zero property that provides an example of graphs with the maximal distance (originally, it is defined in terms of similarity).

**5. *Zero property*:** the dissimilarity (or the distance) between an empty graph $G_0$ and a clique $K_n$ should be maximal, *i.e.*, $\lim\limits_{n \to \infty} sim(G_0, K_n) = 0$.

For some applications, one can extend the property and claim that distance function should produce a large value for two complementary graphs.

In *Koutra, Vogelstein & Faloutsos (2013)* there were also introduced some additional intuitive properties for the graph distance metrics.

**6. *Edge Importance*:** changes that create disconnected components should be penalized more than changes that maintain the connectivity properties of the graphs.

**7. *Weight Awareness*:** in weighted graphs, the bigger the weight of the removed edge is, the greater the impact on the similarity measure should be.

**8. *Edge-"Submodularity"*:** a specific change is more important in a graph with few edges than in a much denser, but equally sized graph.

**9. *Focus Awareness*:** random changes in graphs are less important than targeted changes of the same extent.

In literature, most of the studies on the graph similarity consider only classic axioms 1–4. It is also shown that many models do not satisfy the triangle property. Finally, axioms 5–9 have not been proved for most of the graph distance measures.

# GRAPH DISTANCE MEASURES

There have been developed many models that measure the dissimilarity among the networks. They can be grouped into categories based on different criteria. Such criteria include: information about node labeling (with known/unknown node correspondence), network type (directed/undirected, weighted/unweighted) or the approach that they use (sets comparison, spectral models, graph kernels, *etc*.). For instance, graph similarity models, which are based on node labeling, can be applied to temporal or multiplex networks but cannot be used to compare graphs of different nature. On the contrary, distance measures, which compute graphs statistics, allow to compare any two graphs, however, they do not provide a good performance if node labels are important. In general, the choice of the models depends on the type of the graph and the notion of similarity which better suits the problem under consideration.

It should be noticed that most graph similarity models are usually defined for undirected unweighted graphs, however, some of these models can be extended to directed or weighted graphs. This section provides an overview of these models and discusses their properties and computational complexity.

First, the study presents distance measures that are based on nodes correspondence. This class of models is especially useful in studying temporal or multiplex networks, as they possess similar sets of nodes.

## Distances based on sets comparison

1. *Jaccard index (JI)*

The Jaccard index is one of the simplest instruments for graph comparison that can be applied to all types of networks. It measures the similarity between two graphs as the ratio of intersection and union of edge sets corresponding to two networks, *i.e.*,

$$sim_{JI}(G_1, G_2) = \frac{|E_1 \cap E_2|}{|E_1 \cup E_2|}.$$

This measure can be easily adapted to weighted networks as

$$sim_{WJI}(G_1, G_2) = \frac{\sum\limits_{i,j} \min\left(w_{ij}^{G_1}, w_{ij}^{G_2}\right)}{\sum\limits_{i,j} \max\left(w_{ij}^{G_1}, w_{ij}^{G_2}\right)}.$$

The Jaccard index is the similarity measure that varies from 0 (for distinct graphs) to 1 (for identical graphs). The computational complexity of the Jaccard index is proportional

to the number of edges in two networks. One should note that this index is local as it does not consider the connectivity of the graph or long-distance connections among the nodes.

2. *Graph edit distance (GED)*

The method is discussed in *Sanfeliu & Fu (1983)*, *Bunke et al. (2006)*, *Zeng et al. (2009)* and *Gao et al. (2010)*. It measures the minimum number of graph edit operations to transform one graph $G_1$ to another $G_2$, *i.e.*,

$$d_{GED}(G_1, G_2) = |V_1| + |V_2| - 2|V_1 \cap V_2| + |E_1| + |E_2| - 2|E_1 \cap E_2|.$$

Since the graph edit distance is limited to $|V_1 \cup V_2| + |E_1 \cup E_2|$, there also exists a normalized version of the index

$$d_{nGED}(G_1, G_2) = \frac{|V_1 \cup V_2| - |V_1 \cap V_2| + |E_1 \cup E_2| - |E_1 \cap E_2|}{|V_1 \cup V_2| + |E_1 \cup E_2|}.$$

The graph edit distance is the distance measure that provides 0 for two identical graphs. Note that if two networks have the same set of nodes, the GED measure is equivalent to the Hamming distance while its normalized version is similar to the Jaccard index.

3. *Vertex/Edge overlap (VEO)* (*Papadimitriou, Dasdan & Garcia-Molina, 2008*, *2010*).

The method measures the similarity among the graphs based on the overlap of nodes and edges sets, *i.e.*,

$$sim_{VEO}(G_1, G_2) = 2 \frac{|E_1 \cap E_2| + |V_1 \cap V_2|}{|E_1| + |E_2| + |V_1| + |V_2|}.$$

The VEO measure varies from 0 (for distinct graphs) to 1 (for identical graphs). It is similar to the Jaccard index, however, it also accounts for nodes sets. One should note that models 1–3 are local because they treat all edges equally. In other words, they do not consider whether a particular edge connects two disconnected components or two nodes in a dense network. Moreover, they do not allow comparing graphs of different nature.

4. *k-hop nodes neighborhood (NN)*.

The distance between two graphs can be computed based on the structural equivalence of their nodes. Perhaps the simplest measure of structural equivalence is a count of the number of common neighbors two nodes have (*Newman, 2010*). Thus, the similarity between graphs $G_1$ and $G_2$ can be defined as the average similarity of nodes neighborhood, *i.e.*,

$$sim_{NN}(G_1, G_2) = \frac{1}{|V_1 \cup V_2|} \sum_{v \in V_1 \cup V_2} \frac{|N_1(v) \cap N_2(v)|}{|N_1(v) \cup N_2(v)|},$$

where $N_i(v)$ is a set of neighbors of node $v$ in network $G_i$.

If two networks are similar, the similarity of networks is 1. On the contrary, the distance between two distinct networks is 0. The computational complexity of the model is

proportional to the number of edges. One should also note that this measure is local as it measures only 1-hop distance for each node. Thus, one can extend the measure by considering k-hop neighborhood. This article considers up to 3-hops.

5. *Maximum common subgraph distance (MCS)* (*Bunke et al., 2006*)

The maximum common subgraph distance identifies the maximum common subgraph of two graphs $G_1$ and $G_2$, *i.e.*,

$$d_{MCS}(G_1, G_2) = 1 - \frac{|msc(G_1, G_2)|}{\max(|V_1|, |V_2|)}$$

where $|msc(G_1, G_2)|$ denotes the number of nodes that are presented in the maximum common subgraph. The MCS distance varies from 0 (for identical graphs) to 1 (for distinct graphs). Note that this measure does not distinguish directed and undirected graphs because it relies on nodes sets.

## Distances based on matrix distances

Next, the article considers some models that transform the initial matrix of each graph into a new one and then compute the distance between two matrices. Note that most of these models are not scalable, hence, they cannot be applied to large-scale graphs due to their computational complexity.

6. *Frobenius distance (FRO)*

The Frobenius measure computes the distance between graphs $G_1$ and $G_2$ as the distance between their adjacency matrices, *i.e.*,

$$d_{FRO}(G_1, G_2) = \sqrt{\sum_{ij} \left( a_{ij}^{G_1} - a_{ij}^{G_2} \right)^2}.$$

The Frobenius distance is zero for two identical networks. It is also very similar to the Hamming distance. In general, this measure is bounded by $|V_1 \cup V_2|^2$, thus, it can be additionally normalized to [0,1] interval. Moreover, if graphs $G_1$ and $G_2$ are weighted, one can compute the Frobenius distance between their weighted adjacency matrices $W_1$ and $W_2$.

7. *Vector similarity algorithm (VS)*

The method compares two networks with respect to their edge structure taking into account the importance of nodes (*Papadimitriou, Dasdan & Garcia-Molina, 2008, 2010*). Assume that each node $i$ has a quality score $q_i$, which can be defined externally or defined using the network structure (for instance, by calculating a centrality measure). The one can transform the initial weighted adjacency matrix $W$ of graph $G$ into matrix $\tilde{W} = \left[ \tilde{w}_{ij} \right]$ in order to capture the relative importance of an edge $(i, j)$ to node $i$, *i.e.*,

$$\tilde{w}_{ij} = \frac{q_i w_{ij}}{\sum\limits_{k} w_{ik}}.$$

In other words, the quality score $q_i$ is distributed among all the edges from node $i$. Then the similarity between two graphs $G$ and $G'$ is calculated as

$$sim_{VS}(G_1, G_2) = 1 - \frac{1}{|E_1 \cup E_2|} \sum \frac{\left| \tilde{w}_{ij}^{G_1} - \tilde{w}_{ij}^{G_2} \right|}{\max\left( \tilde{w}_{ij}^{G_1}, \tilde{w}_{ij}^{G_2} \right)}.$$

If two networks are identical, the similarity of networks is 1. On the contrary, the distance between two distinct networks is 0. Note that VS measure is local, however, it may capture the global structure of the network if a quality score is defined using the network structure. In this article, the author uses the PageRank algorithm, which computes the probability of being visited for each node by a random walker, as the quality score (*Brin & Page, 1998*).

### 8. DELTACON

The distance between two graphs can be computed based on nodes affinities (*Koutra, Vogelstein & Faloutsos, 2013*). There are multiple ways to construct a nodes affinity matrix $S = [s_{ij}]$ which shows the affinity of node $i$ to $j$ in graph $G$. For instance, one may compare $k$-hop nodes neighborhood, however, this approach does not take into account the distance to each $k$-hop neighbor. Similarly, one may perform a random walk with restarts (an extension of the well-known PageRank algorithm) that measures the probability of visiting other nodes in a network if a random walk restarts at particular node $i$. This measure considers indirect connections among nodes but, unfortunately, it has a high computational complexity. Thus, *Koutra, Vogelstein & Faloutsos (2013)* use the fast belief propagation (FABR) model that transforms the adjacency matrix $A$ into the matrix of nodes affinities, *i.e.*,

$$S = [s_{ij}] = (I + \varepsilon^2 D - \varepsilon A)^{-1},$$

where $I$ is the identity matrix, $D$ is the diagonal degree matrix, $\varepsilon$ is a small constant capturing the influence between neighboring nodes. It was shown in *Koutra, Vogelstein & Faloutsos (2013)* that FABR can be written in the Personalized RWR-like form, however, it has lower computational complexity.

Therefore, the DELTACON model constructs the nodes affinity matrix for each graph and computes the root Euclidian distance between these matrices, *i.e.*,

$$d(G_1, G_2) = \sqrt{\sum_{ij} \left( \sqrt{S_{ij}^{G_1}} - \sqrt{S_{ij}^{G_2}} \right)^2}$$

The choice of the root Euclidian distance is explained by the fact that it detects small changes in a graph and satisfies properties 5–8 from "Properties of Distance Measures" while the Euclidian distance for the FABR model does not satisfy property 5. Additionally, the authors bound the distance measure to the interval (0, 1] as

$$sim(G_1, G_2) = \frac{1}{1 + d(G_1, G_2)}.$$

Additionally, they propose an approximated version of DELTACON algorithm where all the nodes are randomly divided in $g$ groups and the FABR model is applied to measure the affinity all the nodes to each group.

9. *Polynomial dissimilarity (POL)*

Polynomial dissimilarity is a generalization of the Hamming distance that considers only direct neighborhood of nodes based on adjacency matrices $A_1$ and $A_2$. However, one may extend the measure and also include information about $k$-hop neighborhoods of nodes. Since number of neighbors at distance $k$ is defined by the powers of graphs' adjacency matrices $A_1^k$ and $A_2^k$ while more distant nodes contribute less than direct nodes, *Donnat & Holmes (2018)* defines $k$-hop neighborhood information in graph $G$ using the polynomial $P(A)$, *i.e.*,

$$P(A) = \sum_{l=1}^{k} \frac{A^l}{(N-1)^{\propto(l-1)}},$$

where $N$ is the total number of nodes, $\propto$ is an arbitrary weighting factor. As matrix $A$ can be decomposed as $A = Q\Lambda_A Q^T$, one can rewrite $P(A)$ as

$$P(A) = Q\left(\sum_{l=1}^{k} \frac{\Lambda_A^l}{(N-1)^{\propto(l-1)}}\right)Q^T,$$

where $\Lambda_A$ is a diagonal matrix formed from the eigenvalues of $A$ and $Q$ is a square matrix of eigenvectors. As a result, polynomial dissimilarity compares two networks $G_1$ and $G_2$ in terms of the polynomials of their associated adjacency matrices $A_1$ and $A_2$, *i.e.*,

$$d(G_1, G_2) = \frac{1}{N^2}\sqrt{\sum_{ij}\left(P(A_1)_{ij} - P(A_2)_{ij}\right)^2}.$$

10. *Graph diffusion distance (GDD)*

The graph diffusion distance measures the distance based on diffusion process in graphs (*Hammond, Gur & Johnson, 2013*). To describe the diffusion process on a graph $G$, assume some vector $v(t) \in \mathbb{R}^N$, which indicates the value of the quantity at each vertex at time $t$. Then the diffusion process can be defined as

$$v'(t) = -Lv(t),$$

where $L$ is the graph Laplacian of graph $G$.

With initial conditions $v^{(0)}$ at time $t = 0$, this equation has the analytic solution $v(t) = e^{-tL}v^{(0)}$. Thus, $e^{-tL_1}$ and $e^{-tL_2}$ are Laplacian exponential diffusion kernels that simulates the diffusion in networks $G_1$ and $G_2$ for $t$ timestamps.

According to *Hammond, Gur & Johnson (2013)*, the graph diffusion distance defines the distance between graphs as the Frobenius norm between two diffusion kernels at the timestamp $t^*$ where the two kernels are maximally different, *i.e.*,

$$d(G_1, G_2) = \max_t \sqrt{\sum_{i,j} (e^{-tL_1} - e^{-tL_2})_{ij}^2}.$$

### 11. *Resistance perturbation (RP)*

The resistance perturbation compares two networks $G_1$ and $G_2$ with respect to their resistance matrices (*Monnig & Meyer, 2018*). The authors adapt the electrical analogy to network. According to it, the effective resistance between two nodes $u$ and $v$ is the voltage applied between $u$ and $v$ that is required to maintain a unit current through the terminals formed by $u$ and $v$. As a result, the resistance matrix $R = [R_{ij}]_{N \times N}$ for graph $G$ is constructed as

$$R_{ij} = L_{ii}^\dagger + L_{jj}^\dagger - 2L_{ij}^\dagger,$$

where $L^\dagger$ is the Moore-Penrose pseudoinverse of the Laplacian of $G$.

According to *Monnig & Meyer (2018)*, the resistance perturbation is computed as $p$-norm between the two resistance matrices, *i.e.*,

$$d(G_1, G_2) = \left( \left| R_{ij}^{G_1} - R_{ij}^{G_2} \right|^p \right)^{1/p}.$$

If $p \to \infty$, then $d(G_1, G_2) = \max_{i,j} \left| R_{ij}^{G_1} - R_{ij}^{G_2} \right|$. Note that the resistance perturbation distance is not normalized.

## Distances based on nodes/graph statistics

### 12. *Vertex ranking (VR)*

The vertex ranking (VR) compares the two graphs in terms of nodes rankings (*Papadimitriou, Dasdan & Garcia-Molina, 2008, 2010*). In other words, the distance between two graphs is small if the ranking of vertices is similar. The authors adapt the Spearman correlation coefficient to networks and the define the similarity measure as

$$sim(G_1, G_2) = 1 - \frac{2 \sum_{v \in V_1 \cup V_2} w_i \left( \pi_{i,G_1} - \pi_{i,G_2} \right)^2}{D},$$

where $w_i$ is the quality of node $i$ (defined externally or based on the network structure), $\pi_{i,G_1}$ and $\pi_{i,G_2}$ are the rankings of node $i$ in graphs $G_1$ and $G_2$, and $D$ is a normalized coefficient that limits the maximum value of the fraction to 1. This study will use the PageRank score as the quality of each node.

Apart from nodes statistics, one can measure the similarity between two graphs $G_1$ and $G_2$ in terms of graphs statistics. Typical graph statistics may include the degree distribution, the distance distribution, the clustering coefficient distribution, graphlets statistics, *etc.* In other words, each graph is characterized by a set of features or distributions that is used to measure their similarity. Next, the author presents some measures that are based on this approach. Note that these models are invariant to nodes labeling, thus, they allow to compare graphs of different nature or graphs with unknown labels.

### 13. *Degree Jenson–Shannon divergence (degreeJSD)* (*Carpi et al., 2011*)

The degreeJSD model measures the distance between two graphs using Jenson–Shannon divergence between two degree distributions $P_1$ and $P_2$, *i.e.*,

$$d(G_1, G_2) = \sqrt{\mathcal{J}(P_1, P_2)}.$$

### 14. *Portrait divergence (POR)*

The measure is proposed in *Bagrow & Bollt (2019)*. Instead of directly comparing networks $G_1$ and $G_2$, the model computes their portraits $B_1$ and $B_2$ that represent information about number of nodes who have $k$ nodes at distance $l$, $0 \leq k \leq N - 1$, $0 \leq l \leq d$, $d$—diameter of the network. Portraits $B_1$ and $B_2$ are further converted into two distributions $P_1$ and $P_2$, which describe the probability on the two randomly chosen nodes being connected at distance $l$ and having $k$ neighbors for one of them by equation

$$P(k, l) = \frac{kB(k, l)}{\sum\limits_{c} n_c^2},$$

where $n_c$ is the total number of nodes in the component $c$ of graph $G$.

As a result, the portrait divergence is defined using Jenson-Shannon divergence between two probability distributions $P_1$ and $P_2$, that describe distribution for all rows of portraits $B_1$ and $B_2$,

$$d(G_1, G_2) = \mathcal{J}(P_1, P_2).$$

One should note that, similarly to the degreeJSD measure, the portraits divergence takes into account nodes degree distribution, however, it also considers the connectivity of the graph. The computational complexity is proportional to the complexity of the shortest-paths problem.

### 15. *Communicability sequence entropy (CSE)*

Communicability sequence entropy, which is initially designed for unweighted undirected graphs, compares two networks with respect to the communicability among the nodes (*Chen et al., 2018*). First, it transforms the initial adjacency matrix $A$ of graph $G$ into the communicability matrix $C$ based on the number of shortest paths between two nodes of different length, *i.e.*,

$$C = \sum_{k=1}^{\infty} \frac{1}{k!} A^k = e^A = \left[ C_{ij} \right]_{N \times N}.$$

Since matrix $C$ is symmetric, the communicability sequence is formed by flattening the upper triangular of $C$, which is then normalized into [0,1] interval to create the communicability distribution $P$. As a result, the similarity of networks $G_1$ and $G_2$ is computed using Jenson–Shannon divergence between two sorted degree distributions $P_1$ and $P_2$.

### 16. *NetSimile measure*

NetSimile measure, which is proposed in *Berlingerio et al. (2012)*, is based on three steps: feature extraction, feature aggregation, and comparison. First, it computes a pre-defined set of numerical features for each node that capture the topology of the graph. For instance, these features include the total number of neighbors of node $v$, its clustering coefficient, the average number of neighbors of node $v$'s neighbors, their average clustering coefficient, *etc*. Second, such information is transformed into a "signature" vector that contains aggregated information (*e.g.*, median, mean, standard deviation, skewness, kurtosis) of each feature. Finally, two graphs $G_1$ and $G_2$ are compared with respect to the Canberra Distance of "signature" vectors, *i.e.*,

$$d_{Can}(G_1, \ G_2) = \sum_{ij} \frac{\left| s_{ij}^{G_1} - s_{ij}^{G_2} \right|}{\left| s_{ij}^{G_1} \right| + \left| s_{ij}^{G_2} \right|},$$

where $s_{ij}^{G_1}$—is $j$-th component of a "signature" vector of feature $i$ in graph $G_1$.

### 17. *Onion spectrum (OnionS)*

The onion spectrum is based on the idea of $k$-core which is the maximal induced subgraph such that all nodes have degree no less than $k$. To find $k$-core, it is necessary to eliminate sequentially nodes with degree less than $k$ and recalculate the degree of remained nodes. The process of $k$-core decomposition is similar to the peeling of an onion while all nodes can be marked with respect to the elimination stage (layers). Similarly, the onion spectrum of a network is defined as the fraction of all nodes which are found in a given layer of the k-core decomposition (*Hébert-Dufresne, Grochow & Allard, 2016*). Overall, the distance between two graphs is calculated with respect to the onion spectrum of each graph

(*e.g.*, compute Jenson-Shannon divergence between two onion spectrum). One should note that this model can be enriched by considering some additional features like nodes degrees, edges characteristics, *etc*.

18. *dk-series*

Another way of measuring topological structure of graph is discussed in *Orsini et al. (2015)*. It is based on the concept of *dk*-distributions which is defined as a collection of distributions of *G*'s subgraphs of size $d = 0, 1, \ldots, N$ in which nodes are labelled by their degrees in *G*. For instance, *0k*-distribution corresponds to the average degree of *G*, *1k*-distribution is a standard degree distribution, *2k*-distribution computes the number of subgraphs of size two between nodes of degrees $k_1$ and $k_2$, *etc*. As a result, *dk*-distributions are constructed for graphs $G_1$ and $G_2$ and the distance is calculated using the Jensen–Shannon divergence. Note that for $d = 1$ the measure is identical to degreeJSD. In this article, 2k-series is considered.

## Spectral distances

Next, the article presents graph distance models based on spectral properties of two graphs. Contrary to models based on nodes correspondence, these modes are invariant to nodes labelling. Therefore, they are more focused on global structural properties of a graph because node labels do not play any role. Spectral distances can be used to compare graphs of different size and nature.

19. *λ-distances*

The distance between networks can be also computed with respect to their spectrum (*Wilson & Zhu, 2008*). Let $\lambda^G = \left\{ \lambda_1^G, \lambda_2^G, \ldots, \lambda_n^G \right\}$ be the eigenvalues of the matrix that represent graph *G*. There are different ways how the matrix can be defined: adjacency matrix $A$, Laplacian matrix $L = D - A$ where $D$ is the degree matrix, normalized Laplacian matrix $\mathcal{L} = I - D^{-1/2}AD^{-1/2}$, *etc*. Since spectrum of matrix captures topological properties of the network, the distance between the graphs can be computed as

$$d(G_1, G_2) = \sqrt{\sum_i \left( \lambda_i^{G_1} - \lambda_i^{G_2} \right)^2}.$$

Similarly to *Koutra, Vogelstein & Faloutsos (2013)*, the author considers three models that compares the spectrum of adjacency matrix (λ-d Adj.), Laplacian matrix (λ-d Lap.) and normalized Laplacian matrix (λ-d N.L.). In *Banerjee (2012)* eigenvalues are used to construct a spectral density function for each graph while the distance is measured with respect to these distributions. The implementation of the latter method for normalized Laplacian matrix using the JS divergence is denoted by "Lap.JS". Some other spectral distances are described in *Jurman, Visintainer & Furlanello (2011)*.

**20. *Ipsen–Mikhailov (IM) distance.***

The Ipsen–Mikhailov (IM) distance employs spectral density of eigenvalues from Laplacian that provide a powerful invariant characterization of graphs (*Ipsen & Mikhailov, 2002*). The definition of the metric follows the dynamic interpretation of a *N*-nodes network as a N-particles molecules connected by identical elastic strings. This dynamical system is described by a set of differential equations $\ddot{x}_i + \sum_{j=1}^{N} A_{ij}(x_i - x_j) = 0$ for $i = 1, \ldots, N$ where $A$ is an adjacency matrix and $x_i$ is a coordinate of a particle. The vibration frequencies of such a molecule can be described by the spectra of the graph.

The authors introduce the spectral density $\rho(\omega)$ for a graph as a sum of narrow Lorentz distributions

$$\rho(\omega) = C \sum_{k=1}^{N-1} \frac{\gamma}{(\omega - \omega_k) + \gamma^2},$$

where $\omega_k = \sqrt{\lambda_k}$ is vibrational frequencies that are given by the eigenvalues of the Laplacian matrix, $C$ is the normalization constant and $\gamma$ is a bandwidth parameter.

The spectral distance between networks $G_1$ and $G_2$ with densities $\rho_1(\omega)$ and $\rho_2(\omega)$ can be defined as

$$d(G_1, G_2) = \sqrt{\int_0^{+\infty} [\rho_1(\omega) - \rho_2(\omega)]^2 d\omega}.$$

**21. *Hamming–Ipsen–Mikhailov (HIM) distance.***

The HIM distance is based on two features. First, it computes the Hamming distance $H(G_1, G_2)$ that indicates the difference for the edges in both networks (see GED distance). Second, it computes Ipsen–Mikhailov $IM(G_1, G_2)$ distance, which is measured as the square-root of the squared difference of the Laplacian spectrum for each network. As a result, the Hamming-Ipsen-Mikhailov (HIM) distance is computed as

$$HIM(G_1, G_2) = \frac{\sqrt{H(G_1, G_2)^2 + \alpha \cdot IM(G_1, G_2)^2}}{\sqrt{1 + \alpha}},$$

where $\alpha$ is an arbitrary parameter that balances the trade-off between global (IM) and local (H) information in networks. If $\alpha = 0$, the HIM distance is identical to the Hamming distance while $\alpha \to \infty$ approaches the HIM distance to the Ipsen–Mikhailov distance. Note that the HIM distance takes values in $[0, 1]$.

**22. *NetLSD measure***

The Network Laplacian Spectral Descriptor (NetLSD) is proposed in *Tsitsulin et al. (2018)* while the idea is similar to the graph diffusion distance. First, it computes the normalized Laplacian matrix $\mathcal{L} = I - D^{-1/2}AD^{-1/2}$ for each graph. This normalized Laplacian matrix, as opposed to the unnormalized version, has a bounded spectrum and

satisfy some important theoretical properties. Next, the authors consider a heat diffusion process in a network that can be defined using a heat equation

$$v'(t) = -\mathcal{L}v(t).$$

The closed-form solution of this equation is given by the kernel matrix $H_t = e^{-t\mathcal{L}}$ that represents the amount of heat transferred among the nodes at time $t$. By controlling $t$, one can obtain representations of varying degrees of locality ($t \to 0$ for the local and $t \to \infty$ for the global structure of the network). However, as the heat matrix involves pairs of nodes, the NetLSD measure computes a collection of heat traces $tr(H_t)$ at different time $t$. As a result, the distance between the graphs can be measured as the Frobenius norm of heat traces.

### 23. *Non-backtracking spectral distance (NBD)*

Another spectral approach is proposed in *Torres, Suárez-Serrato & Eliassi-Rad (2019)*. The NBD is designed for undirected, unweighted networks and relies on the idea of non-backtracking cycle of a graph which is a closed walk that does not retrace any edges immediately after traversing them. Backtracking edges are not taken into account as they are trivial for undirected graphs while non-backtracking cycles may capture some important topological features of a graph such as hubs and triangles.

The model consists of three steps. First, it transforms graph $G$ into a non-backtracking matrix $B$ that can be interpreted as a transition matrix of a random walker that does not perform backtracks. To simplify the calculations, it also iteratively removes the nodes of degree one from the graph $G$ as it does not change the trace of the non-backtracking matrix $B$. Moreover, for computational efficiency it considers a matrix $B' = \begin{pmatrix} A & I-D \\ I & D \end{pmatrix}$ which share the same eigenvalues as matrix $B$. Second, it computes $r$ largest eigenvalues of a matrix $B'$ which capture information about non-backtracking cycles in a graph. Since matrix $B$ is not symmetric and eigenvalues may be complex, one can identify them as points in $\mathbb{R}^2$ by using their real and imaginary parts as coordinates. Finally, non-backtracking spectral distance between two graphs is computed as the distance (*e.g.*, Euclidean, Wasserstein, Hausdorff, *etc.*) between $r$ largest eigenvalues of corresponding non-backtracking matrices.

The authors also proved that NBD is a pseudo-metric as it may provide zero distance for two distinct graphs. However, it is also mentioned that the proposed model satisfies non-negativity, symmetry, the triangle inequality.

### 24. *Distributional non-backtracking spectral distance (d-NBD)*

The measure is proposed in *Mellor & Grusovin (2019)*. Similarly to NBD, d-NBD is based on non-backtracking matrix $B$ of size $2n \times 2n$. However, instead of comparing eigenvalues, it compares their distribution. After computing $r$ largest eigenvalues of a

matrix $B'$, it rescales them such that they lie exclusively in the disk of radius 1 and then construct the empirical cumulative spectral density function $F(r, \theta)$ as

$$F(r, \theta) = \frac{1}{2n} \sum_{i}^{2n} \mathbb{1}_{\left\{ \left| \widehat{\lambda}_i \right| \leq r \right\}} \mathbb{1}_{\left\{ 0 \leq \arg\left( \widehat{\lambda}_i \right) \leq \theta \right\}},$$

where $r \in [0, 1]$, $\theta \in [0, \pi]$, $\widehat{\lambda}_i$ are rescaled eigenvalues, $\arg\left( \widehat{\lambda}_i \right)$ is the argument of $\widehat{\lambda}_i$ and $\mathbb{1}$ is the indicator function. As a result, the d-NBD measures the distance between graphs $G_1$ and $G_2$ with respect to their spectral densities $F_1$ and $F_2$, *i.e.*,

$$d(G_1, G_2) = \frac{1}{\pi} \left( \int_0^1 \int_0^\pi \left( F_1(r, \theta) - F_2(r, \theta) \right)^2 dr d\theta \right)^{1/2}.$$

In *Mellor & Grusovin (2019)* it is mentioned that d-NBD is also pseudo-metric as it may provide zero distance for two distinct graphs.

25. *Quantum Jensen–Shannon divergence (QJSD)*

The Quantum Jensen-Shannon divergence (QJSD) is a generalization of JS divergence that is proposed in *Majtey, Lamberti & Prato (2005)*. For the networks, *De Domenico et al. (2015)* define it through the square root of the Jensen–Shannon divergence between the eigenvalues of the normalized Laplacian matrix (*De Domenico et al., 2015*; *De Domenico & Biamonte, 2016*).

$$d(G_1, G_2) = \sqrt{S_q\left( \frac{\rho + \sigma}{2} \right) - \frac{1}{2} \left( S_q(\rho) + S_q(\sigma) \right)},$$

where $\rho$ and $\sigma$ are density matrices and $q$ is the order parameter, while $S_q(\rho)$ is computed as

$$S_q(\rho) = \frac{1}{1-q} \log_2 \sum \lambda_k(\rho)^q,$$

where $\lambda_k(\rho) = \dfrac{e^{-\beta \lambda_k(L)}}{\sum e^{-\beta \lambda_l(L)}}$, $\lambda_k(L)$ indicates the $k$th eigenvalue of Laplacian matrix of a network $G_i$, $\beta$ is an arbitrary parameter for diffusion propagator. The QJSD measure is symmetric and takes values in $[0, 1]$. Although the properties of QJSD measure are discussed in multiple studies (see *Lamberti et al. (2008)*; *Briët & Harremoës (2009)*), it is not proved in general that QJSD measure is a distance metric. According to *Carpi et al. (2019)*, the main drawbacks of this measure are the lack of local information and the number of isospectral networks with different topological features.

## Hybrid measures

Next, hybrid graph distance measures are discussed. Contrary to previous models, these models compares two networks with respect to various aspects: statistics of nodes, edges or even the whole network.

26. *Signature similarity (SimHash)* (*Papadimitriou, Dasdan & Garcia-Molina, 2008, 2010*)

The method is based on SimHash algorithm which is applied to document comparison (*Charikar, 2002*). Suppose that each graph is characterized by a set of weighted features $L = \{(t_i, w_i)\}$ where $t_i$ is a token ( = feature) and $w_i$ is its weight. For instance, such features can represent the quality of nodes, edges or the whole network. Features with small weights play a less important role than features with large weights.

Each token can be randomly encoded as a multidimensional binary vector $\vec{a}_i$ of size $b$ using a hash function. Then one can define a vector $\vec{h}_G$ of the graph $G$ as $\sum_{t_i} \vec{a}_i \cdot w_i$ and then transform each entry to 1 if it is positive and to 0 otherwise. Then the signature similarity between graphs $G_1$ and $G_2$ is computed as

$$sim_{SimHash}(G_1, G_2) = 1 - \frac{Hamming\left(\vec{h}_{G_1}, \vec{h}_{G_2}\right)}{b},$$

where $Hamming\left(\vec{h}_{G_1}, \vec{h}_{G_2}\right)$ is the Hamming distance which is defined as the number of positions for which the corresponding entries of binary vectors are different. One should note that if all the features are some graphs statistics, the model becomes invariant to graph labeling. This article follows the original article and considers centrality of nodes (PageRank) and importance of edges with respect to VS algorithm as features of the graph.

### 27. D-measure

In *Schieber et al. (2017)* the authors propose *D*-measure which compares the dissimilarity of two networks with respect to their network's distance distributions, to their node's distances distributions, and to the analysis of the alpha centrality. The key concept is the network node dispersion (NND), which is a measure of the heterogeneity of a graph $G$ in terms of connectivity distances. NND is computed as the normalized Jensen–Shannon divergence of node's distances distributions and the average of the $N$ distributions in graph $G$, *i.e.*,

$$NND(G) = \frac{\mathcal{J}(P_1, \ldots, P_N)}{\log(d+1)} = \frac{\frac{1}{N}\sum_{ij} p_i(j) \log\left(\frac{p_i(j)}{\mu(j)}\right)}{\log(d+1)},$$

where $d$ is the network's diameter, $p_i(j)$ is the fraction of nodes that are connected to node $i$ at distance $j$ and $\mu(j) = \frac{\sum_{i=1}^{N} p_i(j)}{N}$ is the average distribution.

As a result, *D*-measure between graphs $G_1$ and $G_2$ is computed as the difference between the graphs averaged node-distance distributions (network's distance distribution), $\mu_1$ and $\mu_2$ as well as between the α-centrality values of the graphs and their complements, *i.e.*,

$$D(G_1, G_2) = w_1 \sqrt{\frac{\mathcal{J}(\mu_1, \mu_2)}{log2}} + w_2 \left| \sqrt{NND(G_1)} - \sqrt{NND(G_2)} \right|$$

$$+ \frac{w_3}{2} \left( \sqrt{\frac{\mathcal{J}\left(P_{\propto G_1}, P_{\propto G_2}\right)}{log2}} + \sqrt{\frac{\mathcal{J}\left(P_{\propto \bar{G}_1}, P_{\propto \bar{G}_2}\right)}{log2}} \right),$$

where $w_1$, $w_2$ and $w_3$ are arbitrary weights of the terms where $w_1 + w_2 + w_3 = 1$ (by default, $w_1 = w_2 = 0.45$, $w_3 = 0.1$), $P_{\propto G_1}, P_{\propto G_2}, P_{\propto \bar{G}_1}, P_{\propto \bar{G}_2}$ are the distributions of the α-centrality values in graphs $G_1$, $G_2$ and their complements $\bar{G}_1$, $\bar{G}_2$. It is also proved in *Schieber et al. (2017)* that $NND(G) < 1$ for any graph and, consequently, $0 \leq D(G_1, G_2) < 1$. If two graphs are identical, the $D$-measure is equal to 0. However, the authors also mention that $D(G_i, G_j) = 0$ if two graphs have the same graphs distance distribution, the same NND and the same α-centrality vector, which might be true for non-isomorphic networks. Finally, $D$-measure for sparse graphs can be adapted for large-scale sparse graphs if the α-centrality comparison is avoided ($w_3 = 0$).

An adaptation of $D$-measure to weighted networks is discussed in *Jiang et al. (2021)*.

### 28. Layer difference (LD) measure

$D$-measure can be adapted to the networks with the same nodes sets. In *Carpi et al. (2019)* the authors propose a novel approach to assess the similarity of layers in a multiplex network, however, it can be defined to measure the similarity of different graphs as well. The proposed LD measure is based on the node distance distribution $N_i^G$ of node $i$ in graph $G$, which indicates the fraction of nodes that are at distance $d$ (shortest path) from node $i$ and the transition matrix $T_i^G$, which shows the probability of visiting each node in one step from node $i$. Thus, the similarity of node $i$ is computed in terms of their local neighborhood and connectivity paths using the Jensen–Shannon divergence, *i.e.*,

$$\mathcal{D}_i(G_1, G_2) = \frac{\sqrt{\mathcal{J}\left(N_i^{G_1}, N_i^{G_2}\right)} + \sqrt{\mathcal{J}\left(T_i^{G_1}, T_i^{G_2}\right)}}{2\sqrt{log2}}$$

With this definition, $\mathcal{D}_i(G_1, G_2) = 0$ if node $i$ has the identical connectivity paths in networks $G_1$ and $G_2$ while $\mathcal{D}_i(G_1, G_2) = 1$ indicates that node $i$ is not connected (not present) in one graph, while there are paths connecting $i$ to all nodes in the other graph. As a result, the LD measure is computed as the average value of $\mathcal{D}_i(G_1, G_2)$ over all the nodes, *i.e.*,

$$D(G_i, G_j) = \frac{\sum_i \mathcal{D}_i(G_1, G_2)}{|V_i \cup V_j|}.$$

One should note that *Carpi et al., 2019* extended LD measure to assess the diversity in multiplex networks.

### 29. SLRIC similarity

In *Aleskerov & Shvydun (2019)* the authors propose a model that evaluates the distance between two networks in terms of their structure and sets of central elements. In other words, two graphs are similar if they have comparable topological structure as well as the set of the most important nodes. Thus, the SLRIC similarity measure is designed for complex systems where nodes influence is essential for the analysis of the network evolution.

To compare the centralities of nodes, the authors adapt an idea of interval orders (proposed in *Wiener (1914)*) and construct a matrix $R = \left[ r_{ij} \right]$ for each graph $G$, which represents information about relative ranking of nodes, as

$$r_{ij} = \begin{cases} 1, & c_i - c_j > \varepsilon, \\ 0, & otherwise, \end{cases}$$

where $c_i$ and $c_j$ is a centrality of nodes $i$ and $j$, $\varepsilon \geq 0$ is a pre-defined constant which can be set to deal with inaccuracy of initial data.

The distance between two rankings for networks $G_1$ and $G_2$ can be evaluated as the normalized Hamming distance, *i.e.*,

$$d(R_1, R_2) = \frac{\sum_{j \neq k}^{n} \left| r_{ij}^{G_1} - r_{ij}^{G_2} \right|}{n \cdot (n-1)},$$

where $n$ is the total number of nodes in networks $G_1$ and $G_2$. The distance between two rankings varies from 0 (the same ranking of nodes) to 1 (the opposite ranking of nodes).

Similarly, the distance in terms of the network topology for networks $G_1$ and $G_2$ is calculated as

$$d\left( \tilde{C}_1, \tilde{C}_2 \right) = \frac{\sum_{j,k}^{n} \left| \tilde{c}_{ij}^{G_1} - \tilde{c}_{ij}^{G_2} \right|}{n^2 \cdot \gamma},$$

where $\tilde{C}_1 = \left[ \tilde{c}_{ij}^{G_1} \right]$ and $\tilde{C}_2 = \left[ \tilde{c}_{ij}^{G_2} \right]$ are the matrices of indirect influence of nodes from networks $G_1$ and $G_2$. These matrices can be obtain using SRIC or LRIC modes which are discussed in *Aleskerov, Meshcheryakova & Shvydun (2017)* and *Aleskerov, Shvydun & Meshcheryakova (2021)*. A distinctive feature of these models is that they identify significant edges while insignificant edges are not taken into account. Finally, a single value for the distance between the networks $G_1$ and $G_2$ can be computed as Euclidian norm between two-dimensional vector $\left( d(R_1, R_2), \ d\left( \tilde{C}_1, \tilde{C}_2 \right) \right)$ or according to the formula

$$d(G_1, \ G_2) = \alpha \cdot d(R_1, R_2) + (1 - \alpha) \cdot \ d\left( \tilde{C}_1, \tilde{C}_2 \right),$$

where parameter $\alpha \in [0, 1]$ corresponds to relative importance of the ranking distance. One should note that the properties of the model in *Aleskerov & Shvydun (2019)*, as well as the optimal values of additional parameters, have not been studied in detail.

Table 1 presents a list of selected graph distance/similarity measures and provides their characterization.

In the next Section, 39 graph distance methods are considered: weighted and unweighted versions of Jaccard Index (JI), normalized and unnormalized versions of graph edit distance (GED), vertex/edge overlap (VEO), 1-,2- and 3-hop nodes neighborhood (k-hop NN), maximum common subgraph distance (MCS), normalized, unnormalized and weighted versions of Frobenius distance (FRO), vector similarity algorithm (VS), DELTACON, polynomial dissimilarity (POL), graph diffusion distance (GDD), resistance perturbation (RP), vertex ranking (VR), degree Jenson-Shannon divergence (degreeJSD), portrait divergence (POR), communicability sequence entropy (CSE), NetSimile, onion

**Table 1 Graph distance/similarity measures.**

| № | Name | Idea | Network type | | | | Invariant to node labeling |
|---|------|------|------------|---------|-----------|---------|---|
| | | | Undirected | Directed | Unweighted | Weighted | |
| 1 | Jaccard index (JI) | Sets comparison | + | + | + | + | − |
| 2 | Graph edit distance (GED) | | + | + | + | − | − |
| 3 | Vertex/edge overlap (VEO) | | + | + | + | − | − |
| 4 | k-hop nodes neighborhood (k-hop NN) | | + | + | + | − | − |
| 5 | Maximum common subgraph distance (MCS) | | + | − | + | − | − |
| 6 | Frobenius distance (FRO) | Matrix distances | + | + | + | + | − |
| 7 | Vector similarity algorithm (VS) | | + | + | + | + | − |
| 8 | DELTACON | | + | + | + | + | − |
| 9 | Polynomial dissimilarity (POL) | | + | + | + | − | − |
| 10 | Graph diffusion distance (GDD) | | + | + | + | + | − |
| 11 | Resistance perturbation (RP) | | + | + | + | + | − |
| 12 | Vertex ranking (VR) | Nodes statistics | + | + | + | + | − |
| 13 | Degree Jenson-Shannon divergence (degreeJSD) | Graph statistics | + | + | + | − | + |
| 14 | Portrait divergence (POR) | | + | + | + | − | + |
| 15 | Communicability sequence entropy (CSE) | | + | + | + | − | + |
| 16 | NetSimile measure | | + | + | + | + | + |
| 17 | Onion spectrum (OnionS) | | + | − | + | − | + |
| 18 | dk-series | | + | − | + | − | + |
| 19 | λ-distances | Spectral | + | + | + | − | + |
| 20 | Ipsen–Mikhailov (IM) distance | | + | + | + | − | + |
| 21 | Hamming–Ipsen–Mikhailov (HIM) distance | | + | + | + | − | + |
| 22 | NetLSD measure | | + | + | + | − | + |
| 23 | Non-backtracking spectral distance (NBD) | | + | + | + | − | + |
| 24 | Distributional non-backtracking spectral distance (d-NBD) | | + | + | + | − | + |
| 25 | Quantum Jensen-Shannon divergence (QJSD) | | + | + | + | − | + |
| 26 | Signature similarity (SimHash) | Hybrid | + | + | + | + | ± |
| 27 | D-measure | | + | + | + | + | + |
| 28 | Layer difference (LD) | | + | + | + | − | − |
| 29 | SLRIC similarity | | + | + | + | + | − |

spectrum (OnionS), dk-series (d = 2), 4 versions of λ-distances (λ-d Adj., λ-d Lap., λ-d N. L., Lap.JS), Ipsen–Mikhailov (IM) and Hamming–Ipsen–Mikhailov (HIM) distances, network Laplacian spectral descriptor (NetLSD), non-backtracking spectral distance (NBD) and distributional NBD (d-NBD), quantum Jensen–Shannon divergence (QJSD), signature similarity (SS), D-measure, LD-measure and SLRIC similarity (SLRIC-sim). One should note that unweighted versions of Frobenius distance and graph edit distance are identical by their definition.

# VALIDATION

"Graph Distance Measures" discusses graph distance measures which are based on different notions of similarity. Unfortunately, some models cannot be applied to large networks due to their computational complexity. Therefore, there is a need to compare these models in order to understand their correlation.

"Validation" provides a comparison of the discussed models. First, the author examines small graphs and presents the main difference among the graph distance measures. Second, experiments on synthetic networks are conducted: one can generate a graph and then perform abrupt changes of its structure in order to evaluate the correlation between distance measures. Next, the performance of distance measures on some real temporal networks is compared. Finally, the runtime of graph distance measures is evaluated.

The comparison of the models is performed in Python. Note that some distance measures are computed using the Python software package, netrd (*McCabe et al., 2021*).

## Comparison of the models on small graphs

Let us consider some small graphs in order to understand the difference between the distance measures. To perform such an experiment, the author examines graphs from *Koutra, Vogelstein & Faloutsos (2013)* and *Schieber et al. (2017)*. Note that all these graphs are undirected and unweighted, as many models cannot be applied to weighted/directed networks.

The graphs are presented in Fig. 1. The interpretation of the examples is provided below:

- Example 1 considers a complete graph with five nodes (G1). For G2 and G3, one and two edges have been removed.
- Example 2 examines a circle graph with five nodes (G1). In graph G2 an edge between nodes 4 and 5 is removed. Graph G3 is disconnected.
- Example 3 studies three graphs with the same number of nodes and edges: G1 and G2 are connected while graph G3 contains three identical disconnected components.
- Example 4 considers two complete graphs with five nodes, which are connected by an edge. For graphs G2 and G3, an edge was removed from the graph.
- Example 5 studies three graphs with the same number of nodes and edges. Only graphs G1 and G2 are connected.
- Example 6 examines a complete graph with five nodes which is connected to a chain with four nodes. In graphs G2 and G3 an edge was excluded.

Intuitively, graphs G1 and G2 are more similar than G1 and G3 (test 1). Similarly, the distance between G1 and G2 should be less than the distance between G2 and G3 (test 2). Table 2 provides the results of the experiment. An empty cell in the table denotes tests that could not verified using 'netrd' package. According to the results, there are only two graph distance measures (deltacon, IM) that passed all the tests. CSE, NetSimile, NetLSD passed 11 of 12 tests. As it is expected, most simple models that rely on short connections (JI, GED,VEO, FRO) did not satisfy most of the tests. Interestingly, OnionS provided a

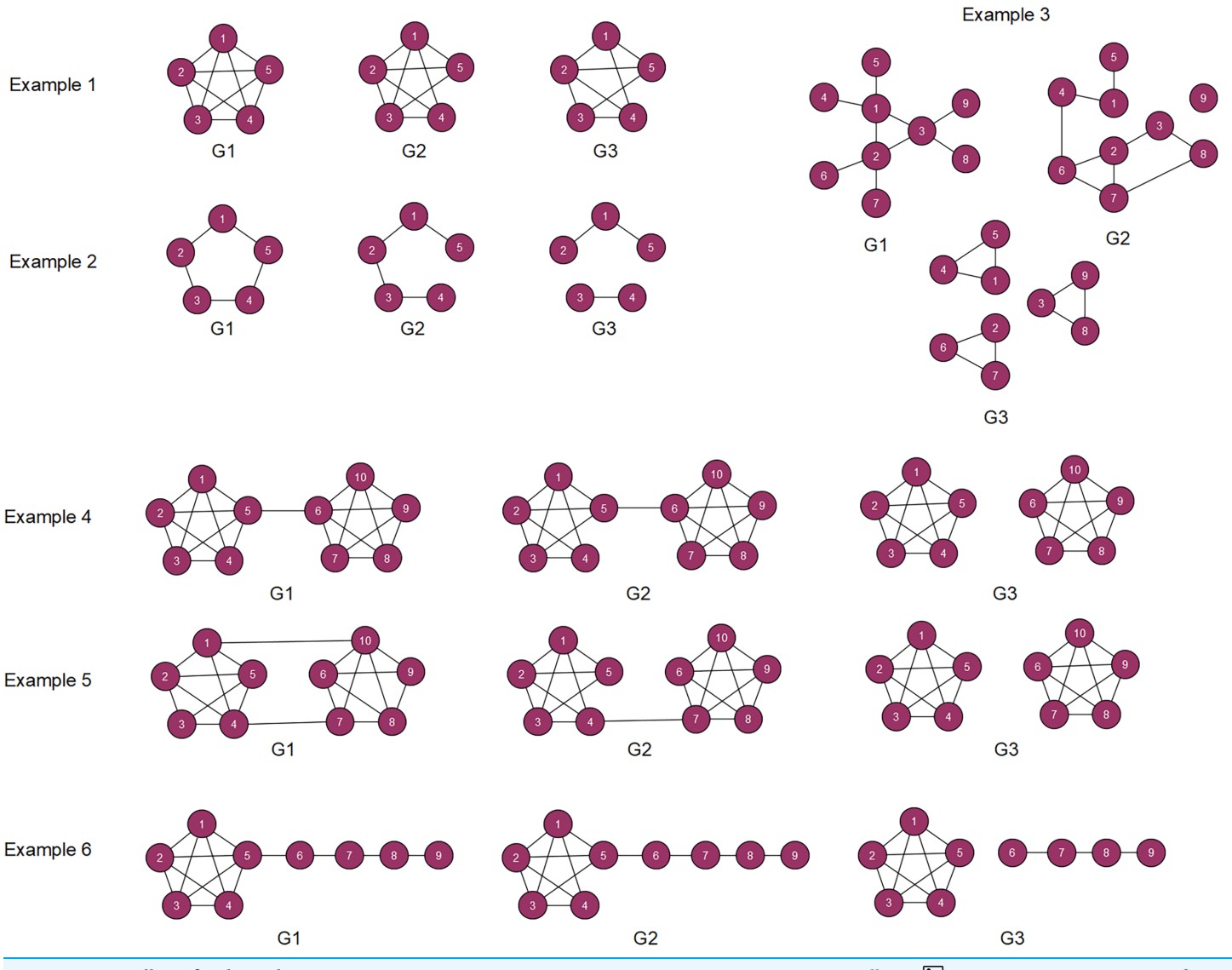

**Figure 1** Small artificial graphs.

poor performance. Furthermore, examples 3 and 5 have the largest amount of tests that graph distance measures did not pass. Overall, it is observed that 21 graph distance measures have passed more than 50% of tests.

## Comparison of the models on artificial time-evolving networks

To compare graph distance measures on more complex structures, the author conducts the following experiment. First, there were generated 1,000 Erdős–Rényi random graphs with $n = 100$ nodes where edges appear with probability $p = 2/(n - 1)$. The value of $p$ is chosen to keep the graph sparse and ensure that most nodes correspond to the giant component. Second, there were performed abrupt changes in the structure of the graph to simulate its evolution. At every iteration some random edges are randomly removed and
Table 2 Comparison of graph distance measures on small graphs.

| Model | Example 1 | | Example 2 | | Example 3 | | Example 4 | | Example 5 | | Example 6 | |
|---|---|---|---|---|---|---|---|---|---|---|---|---|
| | Test 1 | Test 2 | Test 1 | Test 2 | Test 1 | Test 2 | Test 1 | Test 2 | Test 1 | Test 2 | Test 1 | Test 2 |
| JI | + | + | + | + | − | − | − | + | − | − | − | + |
| GED | + | + | + | − | − | − | − | + | − | − | − | + |
| GED norm | + | + | + | + | − | − | − | + | − | − | − | + |
| VEO | + | + | + | + | − | − | − | + | − | − | − | + |
| 1-hop NN | + | − | + | − | − | − | − | + | − | − | + | + |
| 2-hop NN | − | − | + | + | + | + | + | + | + | + | + | + |
| 3-hop NN | − | − | + | + | + | + | + | + | + | + | + | + |
| MCS | − | − | + | + | + | + | + | + | + | + | + | + |
| FRO | + | + | + | − | − | − | − | + | − | − | − | + |
| VS | + | + | + | + | − | + | + | + | − | − | + | + |
| deltacon | + | + | + | + | + | + | + | + | + | + | + | + |
| POL | + | − | + | − | − | − | − | + | − | − | + | + |
| GDD | − | + | + | + | + | + | + | − | − | + | + | + |
| RP | + | + | | | | | | | | | | |
| VR | + | − | + | − | − | + | + | + | + | + | + | + |
| DegreeJSD | + | − | + | − | + | − | + | + | + | + | − | − |
| POR | + | − | + | − | + | − | + | + | + | + | + | + |
| CSE | + | − | + | + | + | + | + | + | + | + | + | + |
| NetSimile | + | + | + | − | + | + | + | + | + | + | + | + |
| OnionS | − | − | − | − | − | − | − | − | + | − | − | + |
| dk-series | + | − | + | + | − | − | + | + | + | + | − | − |
| λ-d Adj. | + | − | + | − | + | + | + | + | + | + | − | + |
| λ-d Lap. | + | − | + | − | + | + | + | + | + | + | − | + |
| λ-d N.L. | + | − | + | − | + | + | + | + | + | + | − | + |
| Lap.JS | + | + | − | − | + | + | − | − | + | − | + | + |
| IM | + | + | + | + | + | + | + | + | + | + | + | + |
| HIM | + | + | + | − | + | − | + | + | − | − | − | + |
| NetLSD | + | + | + | + | + | − | + | + | + | + | + | + |
| NBD | + | − | | | | | + | + | + | − | | |
| d-NBD | + | − | | | + | − | + | + | − | − | − | − |
| QJSD | + | + | + | − | − | − | − | + | − | − | + | + |
| SS | + | + | − | − | + | − | + | + | + | + | + | + |
| d-measure | + | − | | | | | | | | | | |
| LD-measure | + | − | | | | | | | | | | |
| SLRIC-sim | − | − | + | + | − | − | + | + | + | − | + | + |

added to the graph (the total number of iterations is 100). Finally, the author computes various graph distance measures and evaluates the correlation coefficient among them. Note that all these graphs are undirected and unweighted in order to compare all graph distance measures.

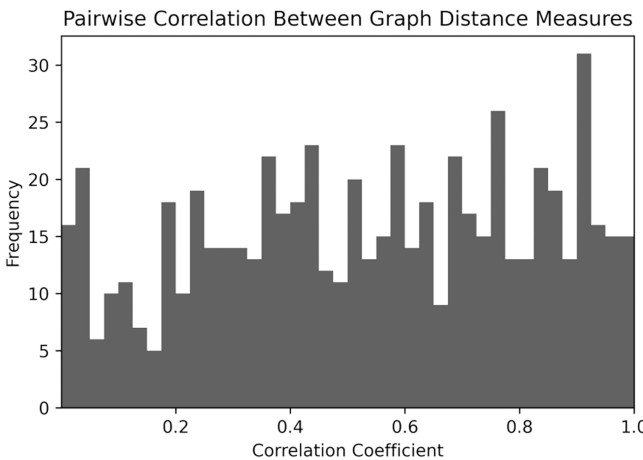

**Figure 2  Histogram over Kendall.**         

Next, the author provide information about Kendall correlation (*Kendall, 1970*), that counts the number of pairwise disagreements between two ranking lists. It is observed that among 630 possible pairs of graph distance measures 144 pairs indicate a very strong correlation (>0.8) and 217 pairs have a weak correlation (<0.4). Overall, it can be concluded that although graph distance measures are based on different concepts of similarity, there exist many measures that have a good correspondence to each other (see Fig. 2). It should be noticed that graph distance measures have been also compared with respect to Spearman and Pearson correlation coefficients, however, these results are not provided in the article because of their high similarity to Kendal correlation.

Since some graph distance measures are strongly correlated to each other, one can combine them in clusters. In this work, the agglomerative clustering algorithm is applied and uses correlation coefficient is used as the distance between different graph similarity models. The results are provided in Fig. 3.

According to Fig. 3, 36 graph distance models have been combined in 12 groups. There are six clusters that contain only one instance: MSC, d-NBD, SS, OnionS, POR, NetSimile. These models did not provide strong correlation with other distance measures on time-evolving random graph models. The remaining six clusters contain eight, six, five, five and four graph distance measures. Some of the clusters include methods that are based on the same idea. For instance, there is a cluster with five spectral methods (λ-d Adj., λ-d Lap., NBD, IM, HIM). There is also a group of methods with a very strong correlation (>0.92) between each other: FRO, QJSD, GED, POL, FRO (norm). Interestingly, the experiments on random graph models demonstrate that models of different nature are well correlated and that some comprehensive methods are agreed well to simple models.

## Comparison of the models on real networks

In this subsection, graph distance measures are compared on the real network that evolves over time. In this work, the author performs the comparison using the crowdsourced air traffic data from the OpenSky Network (*Strohmeier et al., 2021*) during 2019–2021. The choice of the dataset can be explained by the fact that air transportation suffered major

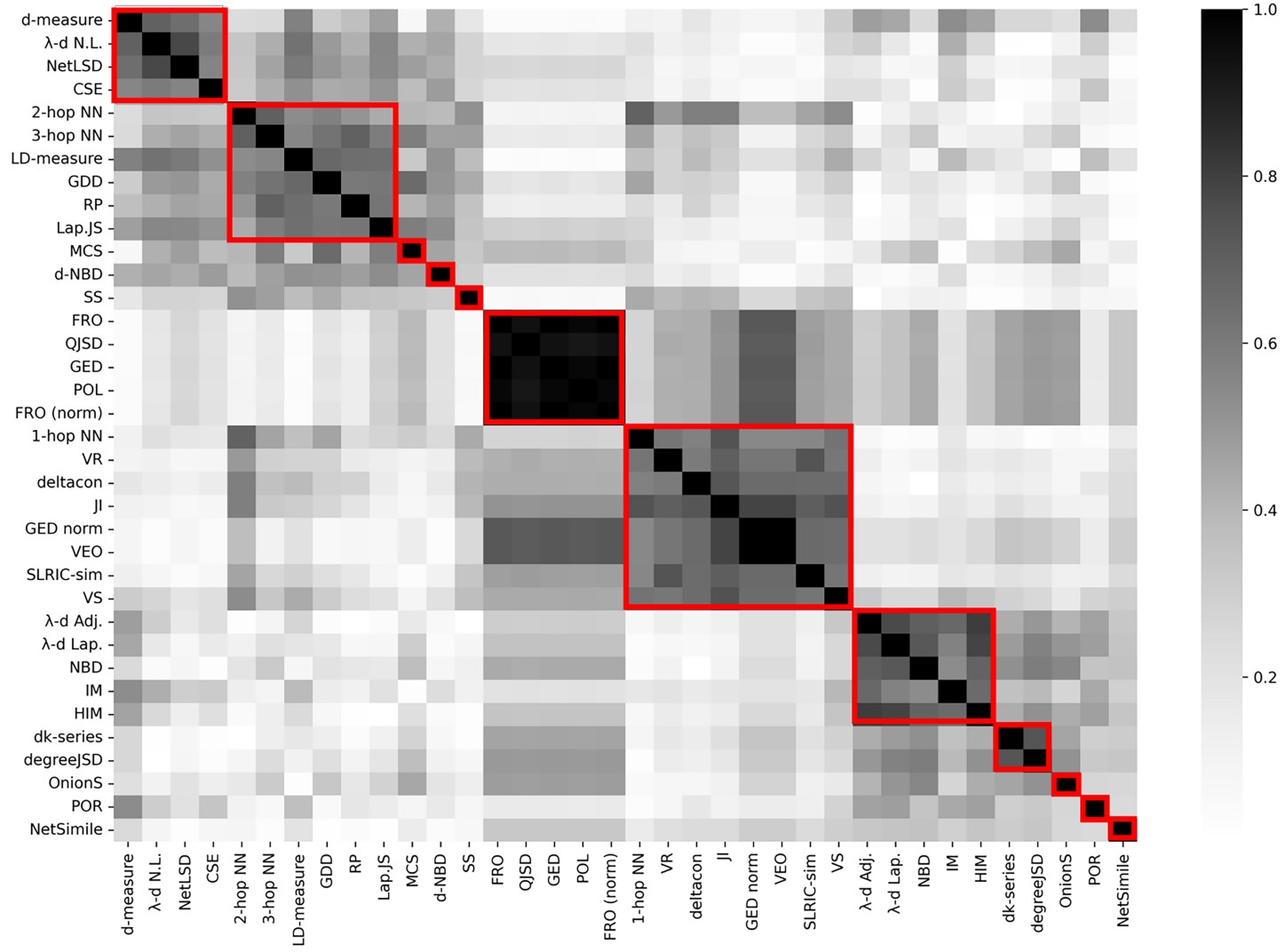

**Figure 3 Correlation matrix (absolute value) between graph distance measures on random graph models.** Red rectangles denote clusters with respect to agglomerative hierarchical clustering algorithm.               

losses during the pandemic and, thus, there were large structural changes in the network due to imposed lockdowns and travel restrictions.

The air traffic data include information about the origin and destination airports for each flight as well as the date of the last airborne message received by the OpenSky Network. It is updated monthly and contains information about 81,954,814 flights seen by the OpenSky Network from January 2019 to December 2021. This article aggregates all the flights into the monthly level in order to construct 36 networks that correspond to air traffic network between countries at a particular month and then perform their pairwise comparison. Each network includes from 93 to 102 nodes while the graph density is relatively dense (0.23–0.31).

**Table 3 Most dissimilar air transportation networks.**

| Network 1 | Networks 2 | Graph distance measure (Total) |
| --- | --- | --- |
| 2020-04 | 2021-06 | JI, GED, GED norm, 1-hop NN (5) |
| | 2021-07 | VS, IM, λ-d Adj., NBD (4) |
| | 2019-10 | λ-d Lap. (1) |
| | 2021-05 | NetSimile (1) |
| | 2021-08 | GDD (1) |
| | 2021-04 | CSE (1) |
| | 2020-01 | POR (1) |
| | 2021-12 | d-NBD (1) |
| 2019-07 | 2021-06 | FRO, FRO (norm), HIM, POL (4) |
| 2019-02 | 2020-10 | 2-hop NN, SLRIC-sim (2) |
| | 2020-05 | SS (1) |
| 2019-10 | 2019-02 | FRO weighted (1) |
| | 2021-07 | dk-series (1) |
| 2021-05 | 2020-05 | NetLSD (1) |
| | 2019-04 | DegreeJSD (1) |
| 2020-08 | 2021-08 | QJSD (1) |
| | 2021-04 | Deltacon (1) |
| 2019-01 | 2020-10 | 3-hop NN (1) |
| | 2021-01 | MSC(1) |
| 2021-06 | 2019-09 | Lap.JS (1) |
| | 2021-03 | VR (1) |

According to the air traffic data, the largest decrease in the total number of edges occurred in April 2020 (>25% drop). It can be explained by the spread of the pandemic (the World Health Organization declares COVID-19 a pandemic in March 2020) that led to subsequent lockdowns and travel restrictions. Thus, one can suppose that the largest dissimilarity should be between air transportation network of April 2020 and the network of some other period. Interestingly, only 17 graph distance measures consider the network of April 2020 as one of the most dissimilar networks while other measures mark June 2021 (11), July 2021 (five) and some other periods. More details are provided in Table 3.

Finally, information about Kendall correlation between graph distance measures in provided in Fig. 4. Similarly to the previous experiments on random graphs, it can be observed that some graph distance measures can be grouped in clusters because of the strong correlation between them. Overall, the results have a good correspondence with the previous experiments on random graphs. However, due to dense and weighted structure of the network, some models formed separate clusters (e.g., deltacon) or moved to another groups (e.g., 2-hop NN, 3-hop NN).

## Performance of the graph similarity models

An important aspect of the graph similarity models is their scalability. Many real networks may contain billions of nodes and edges and are evolving almost instantly. Thus, there is a

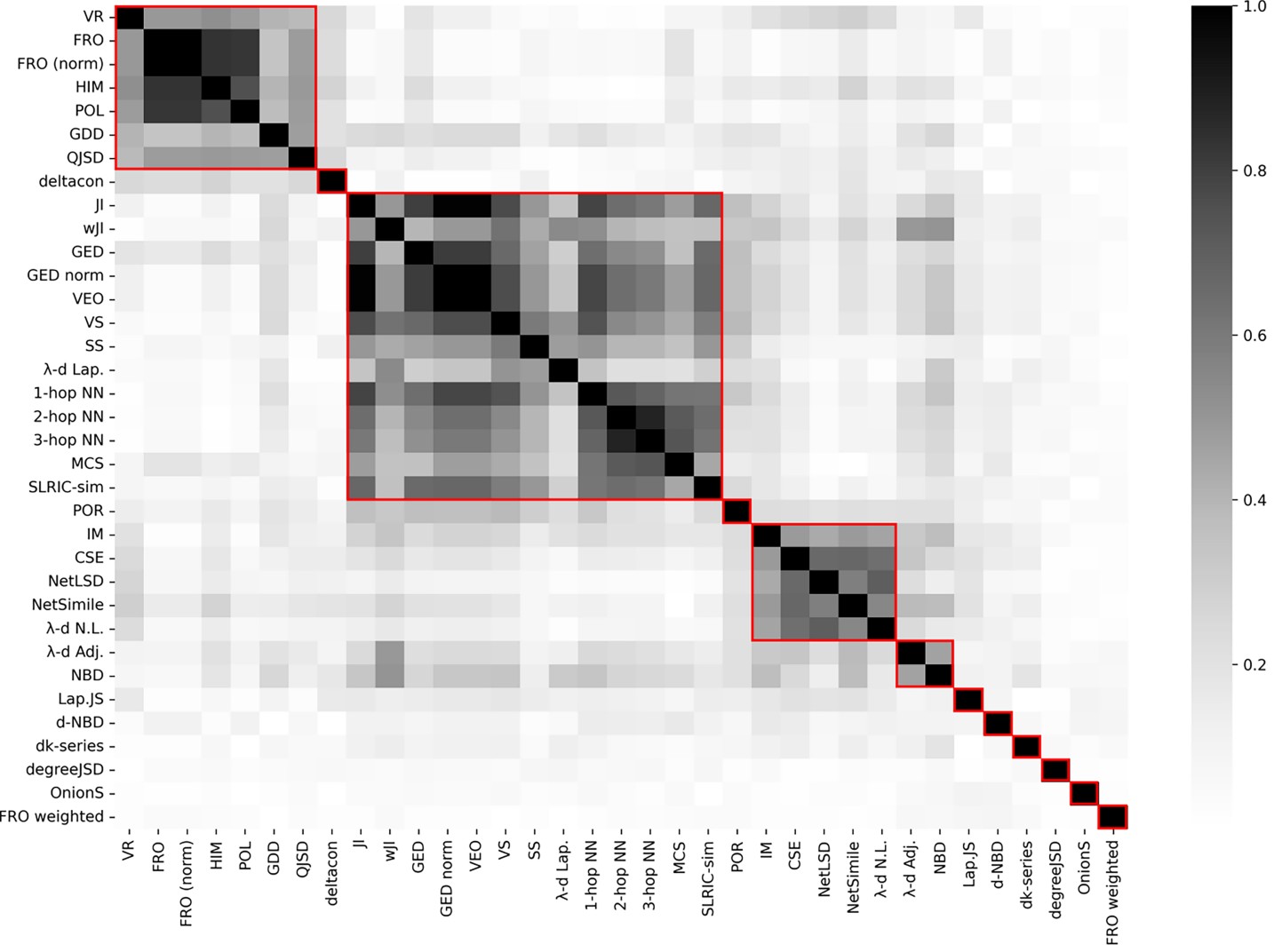

**Figure 4 Correlation matrix (absolute value) between graph distance measures on air traffic data.** Red rectangles denote clusters with respect to agglomerative hierarchical clustering algorithm.

need to find an appropriate measure that is applicable to large networks and captures global information about network.

To compare the average runtime of the models, there have been generated various graph structures. The experiment is performed on Erdős-Rényi random graphs of the same density ($p = 0.05$) that contain from 100 to 3,000 nodes. Due to the computational complexity, the runtime of each model is averaged across 500 iterations for graphs with less than 1,000 nodes and across 10 iterations for larger graphs. It should be mentioned that the article considers existing Python implementations of the methods, however, in theory, some of these algorithms can be further optimized.

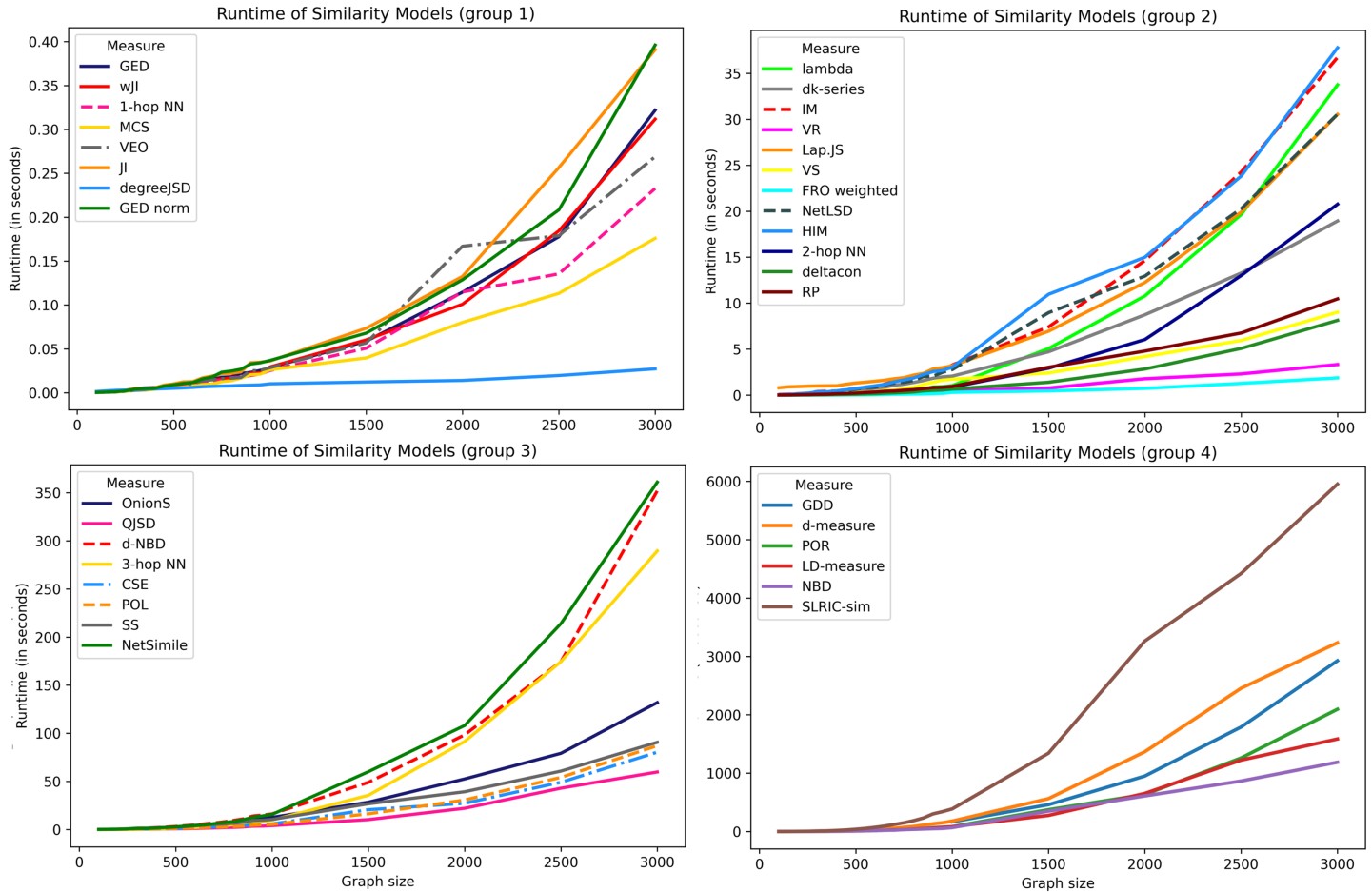

**Figure 5 The runtime of graph similarity models on random networks.** The x-axis corresponds to the number of nodes in the network, and the y-axis corresponds to the average runtime (in seconds).

The results are provided in Fig. 5. For a better visualization, the graph distance measures have been divided into four groups with respect to their runtime. Group 1 (top left) contains measures with a low computational complexity (<0.4 s on graphs with 3,000 nodes). These graph distance measures can be applied to large graphs; however, they consider only the local structure of the network such as direct links. Group 2 (top right) contains measures with relatively low computational complexity (<40 s on graphs with 3,000 nodes), thus, they can be applied to networks with thousands of nodes. The group includes some local measures (*e.g.*, Frobenius distance) but there are also some promising models that employs indirect connections (*e.g.*, deltacon, NetLSD, 2-hop NN) or graphs eigenvalues (*e.g.*, Lap.JS, λ-d Adj.). Next, the graph similarity measures in group 3 (bottom left) consider the global structure of the network, however, their application is limited to relatively small networks with hundreds of nodes. Finally, group 4 (bottom right) includes graph distance measures that can be applied to small networks (<500 nodes) as they are based on the nodes pairwise comparison (*e.g.*, SLRIC-sim) or the shortest paths computation (*e.g.*, D-measure).

# CONCLUSION

Graph similarity is one of the most important problems that can gain a comprehensive insight into network evolution and identify structural phase transitions in graphs or detect anomalies in real systems. As evaluating graph similarity is an ill-defined problem, there has been proposed numerous approaches to measure the distance between graphs. Therefore, it is important to perform the comparison of such models and give guidance on their use.

The article presents established views of the main aspects of the problem and provided a comprehensive comparison of 39 graph distance measures on simple graphs, random graph models and real networks. To the best of the author's knowledge, this is the first study that analyzes such a large set of models.

The graph distance measures have been grouped into six categories based on the underlying idea of computing the distance: sets comparison, matrix distances, nodes statistics, graph statistics, spectral distances and hybrid measures. Models, which are based on the comparison of sets, matrices or node statistics, are mostly designed for temporal and multilayer networks. On the contrary, similarity models, which computes graph statistics or graph spectrum, are invariant to nodes labelling and allow to compare graphs of different size and nature. Furthermore, most of the models can be applied to both directed and undirected graphs but only a few of them can be adapted to weighted graphs.

The comparison of the models on small graphs has shown that deltacon, IM, CSE, NetSimile and NetLSD provide the best correspondence to the intuitive understanding of what graphs are more similar. Although this analysis is rather subjective (there might be as well other examples of graphs), it might offer some insights into the weaknesses of the models.

The experiments on random graphs and real networks demonstrate that graph similarity measures of different nature are surprisingly well correlated and well agreed with simple models. For instance, the polynomial dissimilarity (POL) has a strong correlation with Frobenius distance both on random and real graphs. The Quantum Jensen–Shannon divergence (QJSD) strongly correlates with Frobenius distance on random graphs. The Hamming–Ipsen–Mikhailov (HIM) distance has a good correspondence to other spectral methods. It is also observed that all models based on sets comparison are well correlated with each other. As a result, some comprehensive models can be substituted by simpler ones on large graphs. In addition, the observations can be used as the basis for further comparison of the models.

The graph distance measures have been divided into four groups with respect to their performance. Models that capture the global structure of the network have a larger computational complexity compared to other models that consider only direct connections. However, the key finding is that there exist some models (*e.g.*, NetLSD, deltacon, HIM and some spectral methods) that capture the global structure of the network and provide the output in a reasonable time. Finally, the author provides an implementation in Python of all graph similarity measures that are discussed in the article The Python code is also available on GitHub: https://github.com/SergSHV/.

It is necessary to point out that the article is no aimed to identify the best graph similarity model as the definition of similarity is not well defined and there is no single benchmark to evaluate the models. However, it is strongly believed that the results of the study can be used for the choice of appropriate graph similarity measure and for further development of new models.

## ACKNOWLEDGEMENTS

The author is grateful to Professor F. Aleskerov and anonymous reviewers for their useful comments and suggestions, which helped the author to improve the article.

### Funding

The work of Sergey Shvydun was supported by the Basic Research Program at the National Research University Higher School of Economics (HSE University). The funders had no role in study design, data collection and analysis, decision to publish, or preparation of the manuscript.

### Grant Disclosures

The following grant information was disclosed by the authors:
Basic Research Program at the National Research University Higher School of Economics.

### Competing Interests

The author declares that they have no competing interests.

### Author Contributions

- Sergey Shvydun conceived and designed the experiments, performed the experiments, analyzed the data, performed the computation work, prepared figures and/or tables, authored or reviewed drafts of the article, and approved the final draft.

### Data Availability

The raw measurements for validation are available in the Supplemental Files. The Python code is available on GitHub (https://github.com/SergSHV/).

### Supplemental Information

Supplemental information for this article can be found online at http://dx.doi.org/10.7717/peerj-cs.1371#supplemental-information.

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
