# Peer review of "Models of similarity in complex networks"

_PeerJ Computer Science, doi:10.7717/peerj-cs.1371_

## Round 0.1 · original submission · Major Revisions

Please revise the manuscript according to the reviewers' comments and upload the revised file. Any revisions should be clearly highlighted, for example using the "Track Changes" function in Microsoft Word, color, etc. so that they are easily visible to the editors and reviewers. Please provide a short cover letter detailing any changes, for the benefit of the editors and reviewers.

Reviewer 1 ·

Basic reporting

a) The English language should be improved. Some examples where the language could be improved include lines 163-164 (the same sentence as in the introduction), 203-205, 212 . I suggest seeking the help of a fluent English speaker or contacting a professional editing service.
b) The structure of the article does not conform to the regular format of 'standard sections'. I suggest using the standard format for better clarity and fluency of reading. In addition, Sections, sub-sections and sub-sub-sections must be numerated. Kindly maintain a clear and consistent format throughout the document.
c) In the section "Distance measures and their properties" (line 109), the author uses the term "neighbors" for the first time. This term must be defined before it is used. I suggest including it as a synonym for adjacent nodes in the definition of adjacency between nodes.
d) In the section "Distance measures and their properties", the author must specify the type of graphs considered. Are they simple or multiple ? i.e., can they have several edges between two nodes ?
e) In the section "Distance measures and their properties", the definition of the distance measure is not well structured. I suggest to start by mentioning the purpose of distance measurement, specify that there are several ways to perform this measurement, and then list the different measurement approaches proposed in the literature.
f) In the section "Distance measures and their properties", the author talks about collecting an empirical probability distribution or a feature vector from each graph. The author must clarify : how is this collection done? the empirical probability distribution and the feature vector correspond to what in the considered graphs? This is a very important point to detail and clarify before going to the overview of the different methods, quoted in this part, for the computation of distance between vectors.

Experimental design

The author made a comparison of the different models on various graphs/networks. A very important metric to consider in a such comparison is the algorithmic complexity of the models. However, this metric has not been considered by the author although it is primordial to determine if a model is applicable on large networks and also if it is adapted to real situations where the execution time must be reasonable. I invite the author to take this parameter into consideration to complete the comparison of the models.

Validity of the findings

The conclusion mainly restates what has been mentioned in the abstract and in the introduction. To conclude, it is necessary to recall the context of the study and its importance, and to highlight the different results and findings demonstrated in the study. I suggest to rewrite the conclusion and to state the best model (among those studied) that is the most adapted to real situations of different sizes (small or large).

Reviewer 2 ·

Basic reporting

1. In abstract, add the major finding of the work. It is not clear, what is the major contribution.
2. Highlights the research gaps, major finding and motivation in a separate section of the introduction.
3. In section 2, give the proper citation to the distance measure definition and equation.
4. Discuss the results obtain in more details and their benefits.

Experimental design

.

Validity of the findings

.

Reviewer 3 ·

Basic reporting

the paper is clear and written on very good professional level - English is very good and easy to understand

author refers to respected authors - references are in sufficient number

paper structure is very good, and additional figures and tables make it understandable

raw data are prepared to enable the reviewer to check them - raw data are a very good support to prepare the review

formal results include clear definitions

Experimental design

it is the primary research of the author

the author describes his own research work

all methods are described with sufficient detail

Validity of the findings

the author states his conclusions very well - they are linked to his original research

the author provided all data necessary to check the paper originality

Additional comments

There is a description of compared methods from page 101 to page 662. The paper consists of 896 pages. There are references from page 777 to page 896.

The description of the author's own research is very short. Is it possible to shorten section 2 and add more text to section 3 (Validation)?

---

## Round 0.2 · accepted · Accept

Dear author,

Thank you for submitting your research to PeerJ Computer Science. After finishing the review process we agree that your manuscript is ready for publication.